# Neuroprotective Effect of Piracetam against Cocaine-Induced Neuro Epigenetic Modification of DNA Methylation in Astrocytes

**DOI:** 10.3390/brainsci10090611

**Published:** 2020-09-05

**Authors:** Kalaiselvi Sivalingam, Thangavel Samikkannu

**Affiliations:** Department of Pharmaceutical Sciences, Irma Lerma Rangel College of Pharmacy, Texas A&M University, 1010 W Avenue B, Kingsville, TX 78363, USA; sivalingam@pharmacy.tamhsc.edu

**Keywords:** cocaine, piracetam, DNMTs, TET and astrocytes

## Abstract

Cocaine abuse is known to alter mitochondrial biogenesis and induce epigenetic modification linked with neuronal dysfunction. Cocaine-induced epigenetic modification of DNA methylation and the mitochondrial genome may affect mitochondrial DNA (mtDNA) and nuclear DNA (nDNA), as epigenetic DNA methylation is key to maintaining genomic integrity in the central nervous system (CNS). However, the impact of cocaine-mediated epigenetic changes in astrocytes has not yet been elucidated. In this study, we explored the neuroprotective effect of piracetam against cocaine-induced epigenetic changes in DNA methylation in astrocytes. To study our hypothesis, we exposed human astrocytes to cocaine alone or in combination with the nootropic drug piracetam. We examined the expression of the DNA methyltransferases (DNMTs) DNMT-1, DNMT-3A, and DNMT-3B; global DNA methylation levels of 5-methycytosine (5-mC); and induction of ten–eleven translocation (TET) enzymes in astrocytes. In addition, we analyzed mtDNA methylation by targeted next-generation bisulfite sequencing. Our data provide evidence that cocaine impairs DNMT activity and thereby has impacts on mtDNA, which might contribute to the neurodegeneration observed in cocaine users. These effects might be at least partially prevented by piracetam, allowing neuronal function to be maintained.

## 1. Introduction

Drug abuse and addiction are devastating health conditions in modern society. In the United States, an estimated 8.5 million people suffer from a mental health disorder, a substance use disorder or both [1,2]. The diverse actions of psychostimulants affect intracellular signalling and can alter cellular function, which leads to the behavioural and psychological abnormalities underlying addiction syndrome [3]. These long-lasting neuroadaptations are involved in stable epigenetic modifications to DNA in the central nervous system (CNS), which influences gene regulation through DNA methylation and posttranslational modification, including methylation, phosphorylation, acetylation and deacetylation, of histone proteins [4,5,6]. Several studies have also suggested that epigenetic modifications, including DNA methylation, are critical regulators of gene expression in the CNS.

DNA methylation and its role in gene transcription silencing, of which *DNA m*ethyl*t*ransferases (DNMTs) are important regulators, have been the main focus of epigenetic studies [7]. DNMT-3A and DNMT-3B enzymes exhibit *de novo* methylation during early development [8], while DNMT-1 maintains DNA methylation during mitosis by faithfully propagating symmetrically methylated cytosine-guanine (CpG) sites through associations with the DNA replication machinery [9]. However, all three DNMTs are required for methylation maintenance and embryonic development [10,11]. In specific regulatory regions, DNA methylation plays a crucial role in silencing gene expression by either providing targets for enzymes that modify chromatin to maintain it in a closed and inactive configuration [12] or by preventing the interaction between transcription factors and regulatory regions of the gene [13,14,15]. In addition, DNMTs imply a different regulatory mechanism of gene expression through the DNA methylation process in mitochondrial genome [16]. However, an insight of the mechanistic aspects of the epigenetic modifications in mitochondrial genome is still unclear.

Furthermore, DNA demethylation is regulated by the ten–eleven translocation (TET) enzymes TET1-3, which help to convert 5-methylcytosine (5-mC) to 5-hydroxymethylcytosine (5-hmC) [17,18] DNA demethylation involves the direct removal of the methyl group from 5-mC, base excision of 5-mC by DNA glycosylases, and nucleotide excision of 5-mC by endonucleases [19]. Global and site-specific changes in DNA methylation are observed in addiction [20]. Addiction is associated with abnormal patterns of gene expression in different brain regions, including the nucleus accumbens (NAc), prefrontal cortex, amygdala, and ventral tegmental area [5,20]. The role of DNMT and TET interactions in the maintenance of epigenetic landscapes remains an area of intense research. Nevertheless, epigenetic modifications induced by psychostimulants, especially cocaine, remain poorly understood.

Thus, we investigated the effect of piracetam on cocaine-mediated epigenetic modifications in astrocytes. The nootropic drug piracetam is a cyclic derivative of the neurotransmitter gamma aminobutyric acid (GABA). It improves neural plasticity and can modulate cognitive impairment in aging and dementia [21]. Furthermore, piracetam reduces neuronal loss and increases the number of hippocampal synapses in alcohol-administered rats [21], exerts neuroprotective effects against lipopolysaccharides (LPS) induced neuroinflammation and apoptosis in a rat model [22], and helps to maintain mitochondrial integrity, permeability, and dynamics in neuronal cells [23]. In the present study, we determine the effect of piracetam on cocaine-mediated epigenetic modulation and neuronal dysfunction in astrocytes.

## 2. Materials and Methods

### 2.1. Cell Culture and Reagents

Piracetam (purity > 99%) was purchased from Sigma-Aldrich (St. Louis, MO, USA). Cell culture reagents were purchased from ScienCell (Carlsbad, CA, USA). Primary antibodies were purchased from Proteintech (Rosemont, IL, USA) and EpiGentek (Farmingdale, NY, USA). Electrophoresis reagents and nitrocellulose membranes were purchased from Bio-Rad (Richmond, CA, USA). All other reagents were purchased from Sigma–Aldrich (St. Louis, MO, USA). All fluorescent secondary antibodies were purchased from Thermo Fisher Scientific (Waltham, MA, USA).

### 2.2. Primary Human Astrocytes

In this study, primary human astrocytes (isolated from the cerebral cortex) were obtained from ScienCell (catalog # 1800, ScienCell, Carlsbad, CA, USA). Cultured cells (1 × 10^6^) were maintained in astrocyte basal medium supplemented with fetal bovine serum (FBS) and antibiotic⁄antimycotic solutions at final concentrations of 2% and 1% (ScienCell, Carlsbad, CA, USA).

### 2.3. Drug Treatment

Piracetam and cocaine were prepared in cell-culture-grade distilled water at the appropriate working concentrations. To investigate the protective effect of piracetam, astrocytes were seeded and allowed to attach for further drug treatment. The control cells exposed to media alone and treatment groups were exposed to piracetam (10 µM) alone, cocaine (1 µM) alone, and co-exposure of piracetam (10 µM) and cocaine (1 µM) for 24 h.

### 2.4. Global DNA Methylation Analysis

Whole genomic DNA was extracted from primary astrocytes (2 × 10^6^) using the Blood and Cultured Cell DNA Extraction Kit (EpiGentek, P-1018, Farmingdale, NY, USA) according to the manufacturer’s protocol. The purity of the extracted DNA was determined by measuring the absorbance with a Nanodrop spectrophotometer (Thermo Scientific, Waltham, MA, USA). Further analysis of global DNA methylation was performed using a Global DNA Methylation (5-mC) ELISA Easy kit (EpiGentek, P-1030, Farmingdale, NY, USA). In this experiment, methylated fractions of DNA from the treatment groups were analyzed by using detection antibodies and quantified by measuring the absorbance with a microplate reader (Thermo Scientific, Waltham, MA, USA). The amount of methylated DNA was proportional to the measured OD intensity.

### 2.5. RNA Extraction and Real-Time Quantitative PCR (qRT-PCR)

Total RNA from primary human astrocytes (2 × 10^6^) was extracted using a Qiagen kit (Invitrogen Life Technologies, Carlsbad, CA, USA) according to the manufacturer’s instructions. The purity of the extracted RNA was determined by measuring the absorbance with a Nanodrop spectrophotometer (Thermo Scientific, Waltham, MA, USA). First-strand cDNA was synthesized from total RNA (1000 ng). cDNA was amplified using specific primers targeting DNMT-1 (Hs00154749_m1), DNMT-3A (Hs01027166_m1), DNMT-3B (Hs00171876_m1), TET-1 (Hs00286756_m1), TET-2 (Hs00758658_m1), and TET-3 (Hs00379125_m1), and β-actin (Hs99999903_m1) (Applied Biosystems, Foster City, CA, USA) was used as an endogenous control gene to quantify real-time PCR amplification under different experimental conditions. The PCR protocol was as follows: initial denaturation at 95 °C for 2 min followed by 40 cycles of cyclic denaturation at 95 °C for 3 s and annealing at 60 °C for 30 s.

Relative mRNA expression was quantitated, and the mean fold change in the expression of each target gene was calculated using the comparative ΔΔCT method (transcript accumulation index, TAI = 2_ΔΔCT) and is expressed as the fold change. In addition, the RNA levels obtained for treated samples were normalized to those obtained for the control (untreated) sample. Three independent experiments were performed in triplicate to ensure reproducibility.

### 2.6. Western Blot Analysis

#### 2.6.1. Total Lysates

Astrocytes (2.5 × 10^6^) were grown up to 70% confluent in 75 cm^2^ flask and exposed to cocaine and/or piracetam for 24 h. After the incubation period, the cells were collected, washed twice in 1× phosphate buffered saline (PBS) and then lysed in ice-cold lysis buffer on ice for 1 h. The cell lysates were then centrifuged for 15 min at 13,000 rpm at 4 °C to collect total lysates.

#### 2.6.2. Isolation of the Nuclear Fraction

The NE-PER Nuclear isolation Kit (Thermo Scientific, Waltham, MA, USA) was used to extract the nuclear fraction of astrocytes from different experimental groups. Briefly, primary astrocytes (5 × 10^6^) were grown and exposed to cocaine and/or piracetam for 24 h. After the incubation period, the cells were trypsinized, washed with PBS, and centrifuged (500× *g* for 5 min), and the pellets were suspended in different nuclear extraction assay buffers according to the manufacturer’s protocol. The extracted nuclear fraction was used to examine the nuclear protein expression levels.

#### 2.6.3. Isolation of Mitochondrial Fraction

The Mitochondrial Isolation Kit (Abcam, Cambridge, MA, USA) was used to isolate mitochondria from cocaine and/or piracetam-treated cells. Briefly, primary astrocytes (5 × 10^6^) were grown and exposed to cocaine and/or piracetam for 24 h. At the end of the experimental time period, the cells were trypsinized and washed with PBS. The collected cells were centrifuged (1000× *g* for 10 min) at room temperature, and the pellet was suspended in a mitochondrial assay buffer. After 2 min of incubation on ice, the cells were homogenized with a glass homogenizer using 20 up–down strokes. According to the manufacturer’s protocol, the supernatant was centrifuged at 5000× *g* for 10 min at 4 °C to collect the final mitochondrial fraction. The pellet was then resuspended in lysis buffer to examine the mitochondrial protein expression levels.

Equal amounts of total, nuclear and mitochondrial proteins were resolved by 4–15% gradient polyacrylamide gel electrophoresis and subsequently transferred to a nitrocellulose membrane. The blot was blocked with 5% nonfat milk tris-buffered saline, 0.1% tween 20 (TBST) at room temperature for 1 h and incubated overnight with DNMT-1 (24206-1-AP), DNMT-3A (20954-1-AP), DNMT-3B (26971-1-AP), TET-1 (A- 1020), TET-2 (A- 1701), TET-3 (A-50520), lamin (10298-1-AP), cytochrome c oxidase (COX)-IV (COX-IV) (11242-1-AP) and glyceraldehyde 3-phosphate dehydrogenase (GAPDH) (10494-1-AP) antibodies prepared in TBST (1:1000) and incubated at 4 °C. After being washed with TBST, the blot was incubated with a peroxidase-conjugated secondary antibody (EpiGentek, Farmingdale, NY, USA) for 1 h. The immunoreactive bands were visualized using chemiluminescence reagent. To ensure equal protein loading, lamin, COX-IV and GAPDH were used as internal controls. Densitometric analysis was carried out using image J digitalizing software (NIH Image Software).

### 2.7. Immunofluorescence Staining

Briefly, astrocytes (2 × 10^4^) cells/mL were grown on multi-chamber slides and exposed to cocaine and/or piracetam for 24 h. At the end of the incubation period, the cells were fixed with 4% paraformaldehyde for 15–30 min at room temperature and then permeabilized with 0.2% Triton X-100 in PBS for 15 min at room temperature. Then, the cells were blocked with 5% normal goat serum for 1 h at room temperature. The appropriate primary antibody (diluted 1: 300) was subsequently added, and the slides were incubated overnight at 4 °C. After being washed with PBS, the cells were incubated with specific secondary antibodies (Thermo Fisher Scientific, Waltham, MA, USA), and the cellular nuclei were stained with 4′6-diamidino-2-phenylindole (DAPI, Vector Laboratories, Burlingame, CA, USA), and the slides were examined with a C1-plus Nikon laser-scanning confocal microscope. The fluorescence intensity (calculate the corrected total cell (CTCF)) was measured by using ImageJ software [24].

### 2.8. Multiplex PCR, Library Preparation and Sequencing for Targeted Next-Gen Bisulfite Sequencing (TNGBS)

All bisulfite-modified DNA samples were amplified using separate multiplex or simplex PCRs. The assay designed using Assay Design Service (ADS) software (v1.0.6, QIAGEN, Hilden, Germany) for genes or regions. The PCRs included 0.5 units of Qiagen HotStarTaq, 0.2 µM primers, and 3 µL of bisulfite-treated DNA in a 20 µL reaction mixture. QIAxcel Advanced System (v1.3.0, QIAGEN, Hilden, Germany) were used to quantify the PCR products. Meanwhile, PCR products from the same sample were pooled and purified using QIAquick PCR Purification Kit columns (Qiagen). Libraries were prepared using a KAPA Library Preparation Kit for Ion Torrent Platforms (cat# KK8310) and Ion Xpress™ Barcode Adapters (Thermo Fisher, Waltham, MA, USA). Next, the libraries were purified using Agencourt AMPure XP beads (Beckman Coulter, Brea, CA, USA) and quantified using the Qiagen QIAxcel Advanced System. The barcoded samples were pooled in an equimolar fashion before template preparation, and enrichment was performed on an Ion Chef™ system (Thermo Fisher, Waltham, MA, USA) using Ion 520™ and Ion 530™ Chef reagents. Next, the enriched, template-positive libraries were sequenced on an Ion S5™ sequencer using Ion 530™ sequencing chips (Thermo Fisher, Waltham, MA, USA).

#### Methylation Calculations

The methylation of mitochondrial DNA levels was calculated in Bismark by dividing the number of methylated reads by the total number of reads. The following formulas were used to calculate the percentage of methylation across the mitochondrial genome for TNGBS data:

% methylation (context) = 100 * methylated Cs (context)/(methylated Cs (context) + unmethylated Cs (context)).

### 2.9. Statistical Analysis

Statistical analysis was performed using GraphPad Prism version 6, (GraphPad Software, San Diego, CA, USA). Differences between the control, piracetam alone, cocaine alone and cocaine combined with piracetam groups were calculated using one-way ANOVA. The values are expressed as the mean ± standard deviation, and the significance level was * *p* < 0.05.

## 3. Results 

### 3.1. Effect of Piracetam on Cocaine-Mediated Global DNA Methylation Modifications

Global DNA hypomethylation has a significant impact on the progression of various neurodegenerative diseases and can induce undesirable gene activation, leading to genomic instability [18,25]. To evaluate global DNA methylation, cells were treated with or without cocaine (1 µM) and piracetam (10 µM) for 24 h. We used the colorimetric method to quantify global DNA methylation (5-mC). The covalent addition of a methyl group to the 5-carbon of CpG dinucleotides results in the formation of 5-mC. We observed that, compared to control treatment (17%), exposure to cocaine decreased the level of 5-mC (13%). Interestingly, the impact of cocaine on global DNA hypomethylation was prevented by piracetam (16%) in astrocytes (Figure 1A). 

### 3.2. Protective Effect of Piracetam Against Cocaine-Induced DNMT Gene Expression in Astrocytes

We examined the impact of cocaine on the mRNA expression of methylation regulators (DNMTs) in astrocytes. We found that cocaine treatment downregulated the mRNA expression of DNMT-1 and DNMT-3A and upregulated the expression of DNMT-3B. Interestingly, co-exposure to piracetam restored the mRNA expression of DNMT-1, DNMT-3A and DNMT-3B to control (Figure 1B–D). These results suggest that the protective effects of piracetam help to prevent cocaine-induced hypomethylation and restore genomic stability in primary astrocytes (Figure 1B–D). 

### 3.3. Effect of Piracetam on Cocaine-Induced DNMTs Protein Expression and Subcellular Localization (Nuclear and Mitochondria) in Astrocytes

Next, we addressed whether cocaine has similar effects on DNMT protein expression in astrocytes. In addition, we evaluated the expression levels of DNMTs in the total, nuclear and mitochondrial fractions. The activities of three DNMTs maintain DNA methylation and de novo methyltransferase synthesis. DNMT-1 activity is highly regulated by conformational changes and interdomain interactions [16]. Here, cocaine impacted the enzymatic activities of DNMTs, leading to alterations in the folding of genomic DNA and changes in the accessibility of transcription factors. Importantly, in eukaryotic cells, genetic material is compressed in two distinct compartments, namely, the nucleus and mitochondria. Nuclear and mitochondrial genome methylation in CpG regions leads to chromatin remodelling, which is generally considered as a marker of transcriptional silencing. Changes in transcription factors impair nuclear and mitochondrial function. Thus, we evaluated the nuclear and mitochondrial subcellular localization of DNMT-1, DNMT-3a and DNMT-3b, which methylate nuclear (nDNA) and mitochondrial DNA (mtDNA) in astrocytes. Exposure to cocaine decreased the protein expression of DNMT-1 and DNMT-3a but increased the expression of DNMT-3b. Co-exposure to cocaine and piracetam (10 µM) restored the protein expression of DNMT-1, DNMT-3a and DNMT-3b to control levels in total lysates (Figure 2A–F) as well as in the nuclear fractions (Figure 2G–L). These results are consistent with DNMT’s mRNA expression.

However, exposure to cocaine increased the protein expression of DNMT-1 and decreased the expression of DNMT-3a and DNMT-3b in the mitochondrial fraction (Figure 3A–F). Interestingly, co-exposure to cocaine and piracetam restored the protein expression of DNMT-1, DNMT-3a and DNMT-3b to control levels in mitochondrial fraction (Figure 3A–F). Further immunostaining analysis showed that DNMT-1 was predominantly located in the nuclei and cytoplasm of primary astrocytes. DNMT-1 is diffusely distributed in the nucleus during interphase and moves to replication foci for the early and mid S-phase in mammalian cells [26,27]. Figure shows that exposure to cocaine decreased DNMT-1 expression in the cytoplasm and nucleus (Figure 3G,H). However, piracetam restored the cytoplasmic and nuclear localization of DNMT-1 in astrocytes (Figure 3G).

We also analyzed the different mtDNA methylation targeted genome regions, including mitochondrial ribonucleotide reductase 1-2 (mt-RNR1-2), nicotinamide adenine dinucleotide dehydrogenase (NADH) subunit 1-6 (ND1-ND6), mitochondrial cytochrome c oxidase 1-3 (mt-CO1-3), mitochondrial adenosine triphosphate 8/6 (mt-ATP8/6), and mitochondrially encoded cytochrome B (mt-CYB) by targeted next-generation bisulfite sequencing. The results showed that cocaine exposure decreased the mtDNA methylation in 80 CpG sites among a total of 239. Meanwhile, 40 CpG sites including mt-RNR1, mt-RNR2, ND1, ND4, ND5, mt-CO1, mt-CO2, mt-ATP6 and mt-CYB were predominate, decreasing the methylation level in the mtDNA of cells treated with cocaine compared to that of control cells (Figure 4A–D). Besides, piracetam co-exposure significantly restored mitochondrial methylation levels in the mt-RNR1, mt-RNR2, ND1, ND4, ND5, mt-CO1, mt-CO2, mt-ATP6, and mt-CYB regions of mtDNA (Figure 4A–D).

### 3.4. Protective Effect of Piracetam Against Cocaine-Induced DNA Demethylation

Growing evidence suggests that DNA methylation is a reversible process. The stable DNA methylation status of a gene reflects a balance between methylation and demethylation [28]. To better understand the biological relevance of cocaine-induced demethylation, we analysed mRNA and protein expression to determine the effects of TET family enzymes on DNA methylation. To evaluate the effect of cocaine on the expression of TET enzymes in astrocytes, we exposed human primary astrocytes to cocaine. We found that cocaine treatment for 24-h downregulated TET-1, TET-2 and TET-3 mRNA (Figure 5A–C) and protein expression (Figure 6A–F). Interestingly, cotreatment with piracetam restored the mRNA (Figure 5A–C) and protein expression of TET-1 and TET-3 to control levels (Figure 6A–F). These results suggest that the protective effects of piracetam help to reverse the cocaine-induced loss of demethylation in primary astrocytes.

Further immunofluorescence analysis confirmed the cytoplasmic and nuclear localization of TET-1 in astrocytes. In addition, we observed the difference in the subcellular localization of TET-1. The localization of the TET-1 protein in the nucleus is associated with the active demethylation process [29]. The results of the present study revealed that TET-1 mRNA and protein expression levels were significantly lower in cocaine-exposed cells than in control cells. Similarly, immunofluorescence analysis showed that exposure to cocaine decreased the expression of TET-1 in the cytoplasm as well as in the nucleus (Figure 6G). Simultaneously, piracetam markedly restored the expression of TET-1 in the cytoplasm and nucleus to control levels (Figure 6G,H). Taken together, these findings suggest that piracetam helps to restore the coordination between the mitochondrial and nuclear epigenomes in astrocytes.

## 4. Discussion

Accumulating evidence has suggested that epigenetic modification plays a crucial role in maintaining the drug-addicted state [4,30]. Addiction has been shown to be an altered DNA methylation signature in animal models, leading to changes in neuronal and brain activity following acute and chronic drug exposure [20,31]. Evidence from mammalian studies suggests that chronic cocaine administration interferes with DNA methylation dynamics and that gene expression in the brain may alter vulnerability to drug abuse [32,33]. However, cocaine-induced epigenetic modification of DNA methylation and cross-talk with mtDNA-mediated impairment in astrocytes is unknown. Recent research has focused on the importance of astrocytes and their contribution in neuroepigenetic regulation. Further understanding of epigenetic regulatory functions in astrocytes under pathophysiological conditions will help to identify the precise role of astrocytes in neurodegeneration and/or neuroprotection. In this study, we explored the protective effect of the nootropic drug piracetam against cocaine-mediated epigenetic modulation in astrocytes for the first time.

Epigenetic modifications, such as DNA methylation, histone modification, and microRNAs, are critical regulators of gene expression changes in the CNS under normal and pathological conditions [34,35]. Previous findings have shown that dysregulation of DNA methylation and DNMTs might be involved in cognition and neurodegenerative disorders [36,37]. Cocaine administration induces neural adaptation through the regulation of gene expression. Kaili Anier et al. [37] reported that acute cocaine treatment upregulates DNMT-3A and DNMT-3B gene expression in the NAc. Furthermore, acute and repeated cocaine treatment leads to the hypermethylation of fosB promoter associated CpG islands in the NAc. Additionally, Mari Urb et al. [38] reported that the expression of DNMT-3b in the postnatal mouse brain is relatively low compared to that of DNMT-1 and DNMT-3a. Interestingly, acute (0.5–3 h) exposure to cocaine reduces the mRNA expression of DNMT-1 and DNMT-3a in the NAc, which indicates that cocaine regulates DNMT mRNA expression in a time-dependent manner [39]. Moreover, it is becoming clearer that epigenetic modifications, including DNA methylation and demethylation, vary between cell types [40,41].

Here, we found that cocaine reduced methyltransferase activity associated with downregulated mRNA and protein expression of DNMT-1 and DNMT-3a and upregulated DNMT-3b expression. Interestingly, in astrocytes, this downregulation was reversed by co-exposure with piracetam, which indicates a crucial role for piracetam in preventing cocaine-associated DNA hypomethylation. We obtained similar results in the nuclear fractions, suggesting that cocaine decreases the transcription of DNMT-1 and DNMT-3a and increases the transcription of DNMT-3b in the nucleus.

For the first time, we showed that DNMTs are present in the mitochondrial fraction of astrocytes, indicating that DNMTs exert regulatory effects on mitochondrial genome methylation. Previous studies have reported that dysregulated expression of DNMT-1 isoform3 affects mitochondrial function, inducing decreases in the mitochondrial membrane potential and ATP production and affecting mitochondrial genome-encoded OXPHOS-related genes [42]. Under oxidative and nutritional stress conditions, the epigenome is affected, resulting in hypomethylation of the mitochondrial genome [42]. Recently, Alessandra Maresca et al. [16] reported that two major cellular energy sensors, AMP-activated protein kinase (AMPK) and mammalian target of rapamycin complex 1 (mTORC1), are associated with a severe DNMT-1 mutation and alter mitochondrial function. Our results confirmed that cocaine decreases the DNA methylation of cytosine residues in the mitochondrial genome, which may cause impaired mitochondrial function and energy deficits in astrocytes. Concurrently, piracetam restores the cocaine-associated DNA hypomethylation in mt-RNR1, mt-RNR2, ND1, ND4, ND5, mt-CO1, mt-CO2, mt-ATP6 and mt-CYB regions of mtDNA.

To date, no studies have focused on the complex, cocaine-induced role of both DNA methylation and demethylation in astrocytes. Several studies have reported that DNMT-3A and DNMT-3B initiate the demethylation process [43,44] and that the deaminase activity of these enzymes is involved in a dynamic demethylation-methylation pathway during gene transcription [43,44]. However, the interactions between DNMTs and TETs in the maintenance of the epigenome remain unknown.

Recent studies have focused on the interplay between DNA methylation/demethylation mechanisms involving 5-mC and 5-hmC. The TET family of dioxygenases consists of TET-1, TET-2, and TET-3, which participate in the conversion of 5-mC to 5-hmC [45,46]. Studies have shown that repeated cocaine administration results in decreased TET-1 expression in the NAc but does not affect the levels of TET-2 or TET-3 [47]. Other studies have also reported that the mRNA levels of TET-1 and TET-2 in the NAc, peripheral blood cells (PBCs) and the cerebellum are decreased after acute treatment with cocaine [39]. Here, our results showed that cocaine decreases the mRNA and protein expression of TET-1, TET-2 and TET-3 in astrocytes. Treatment with piracetam reverses the inhibitory effect of cocaine on demethylation, which may help to reverse DNA demethylation in astrocytes. Identifying the effective modulator of hypo/hypermethylation is key in preventing psychostimulant-induced epigenetic modifications. Interestingly, dietary supplementation and administration of methionine and *S*-adenosyl methionine (SAM) provides methyl group substrates for DNMTs, increasing DNA methylation [48]. Additionally, treatment with DNMT inhibitors such as 5-azacytidine and RG108 leads to hypomethylation in an in vitro model [48,49]. We explored the ability of another effective epigenetic modulator, piracetam, to reverse cocaine-induced hypomethylation in astrocytes. Future research will help to delineate the efficacy of piracetam in preserving mitochondrial and nuclear epigenomic integrity in the CNS.

## 5. Conclusions

In conclusion, we hypothesize that cocaine exposure can impair DNA methylation by suppressing the function of nDNA and mtDNA methyltransferases, thereby mediating the inappropriate activation of different genes and the progression of various neuronal diseases. Co-exposure to piracetam leads to potent neuroprotective effects, thereby preventing these epigenetic changes in astrocytes.

## Figures and Tables

**Figure 1 brainsci-10-00611-f001:**
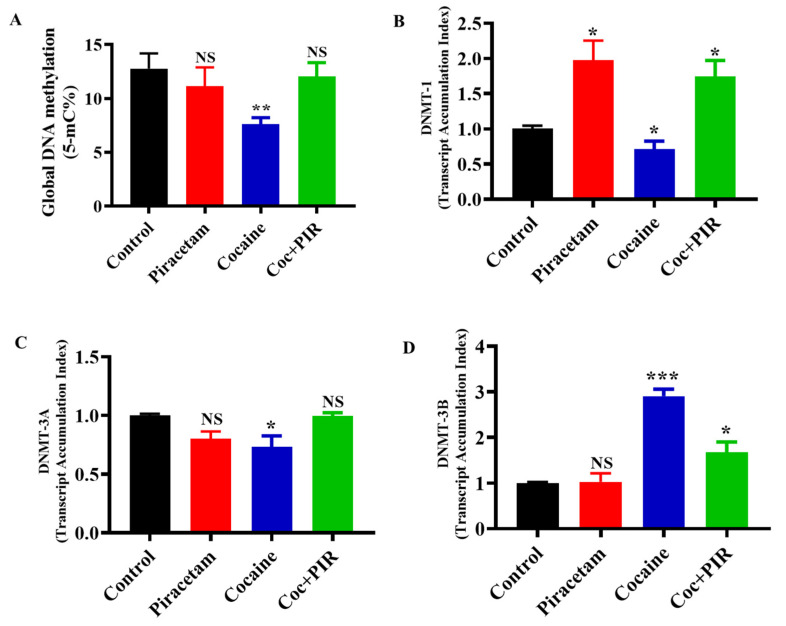
Piracetam reversed the impact of cocaine on global DNA methylation (5-mc) and DNA methyltransferases’ (DNMTs’) gene expression. Human primary astrocytes (2 × 10^6^ cells/mL) were exposed to cocaine (1 µM) and/or piracetam (10 µM) for 24 h. Total cellular DNA were used to determine the global DNA methylation (5-mC) by ELISA (**A**) and the total RNA were analysed (**B**) DNMT-1, (**C**) DNMT-3A and (**D**) DNMT-3B mRNA expression by qRT-PCR. The housekeeping gene β-actin was used as a loading control. The results are expressed as the mean ± SD of the transcript accumulation index (TAI) of three independent experiments. *** *p <* 0.001, ** *p <* 0.01, * *p <* 0.05, NS—nonsignificant.

**Figure 2 brainsci-10-00611-f002:**
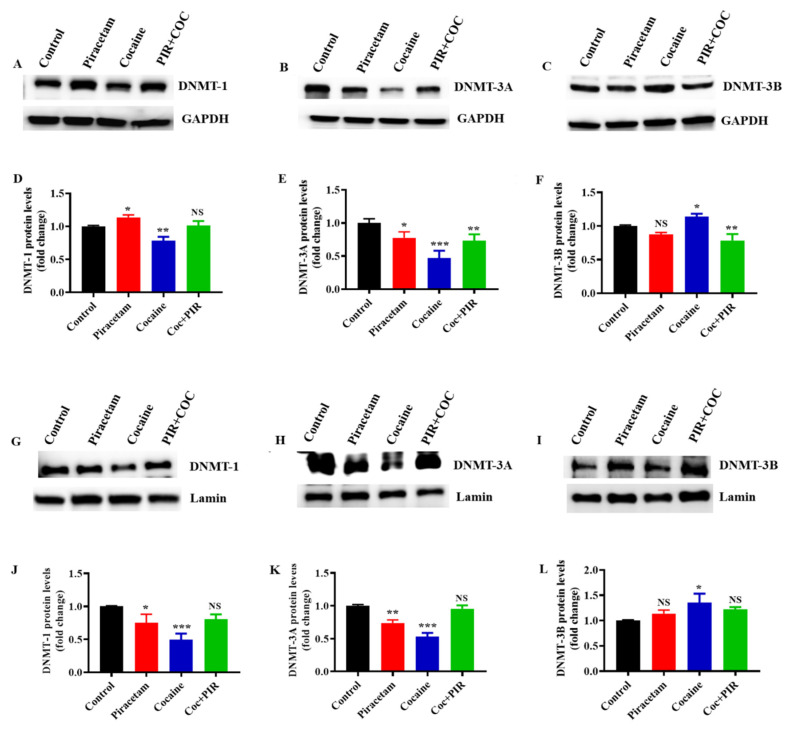
Piracetam reversed the impact of cocaine on DNMT protein expression in primary astrocytes. Human primary astrocytes were exposed to cocaine (1 µM) and/or piracetam (10 µM) for 24 h, and the total and nuclear Fractions were isolated. Representative blots (**A**,**G**) showing the expression of DNMT-1, (**B**,**H**) DNMT-3A and (**C**,**I**) DNMT-3B in the total and nuclear fractions. (**D**–**F**,**J**–**L**) Densitometric analysis of the level of each protein relative to the corresponding level of GAPDH or lamin as a loading control (fold change relative to the control). The data are expressed as the mean ± SD of three independent experiments. *** *p <* 0.001, ** *p <* 0.01, * *p <* 0.05, NS—nonsignificant.

**Figure 3 brainsci-10-00611-f003:**
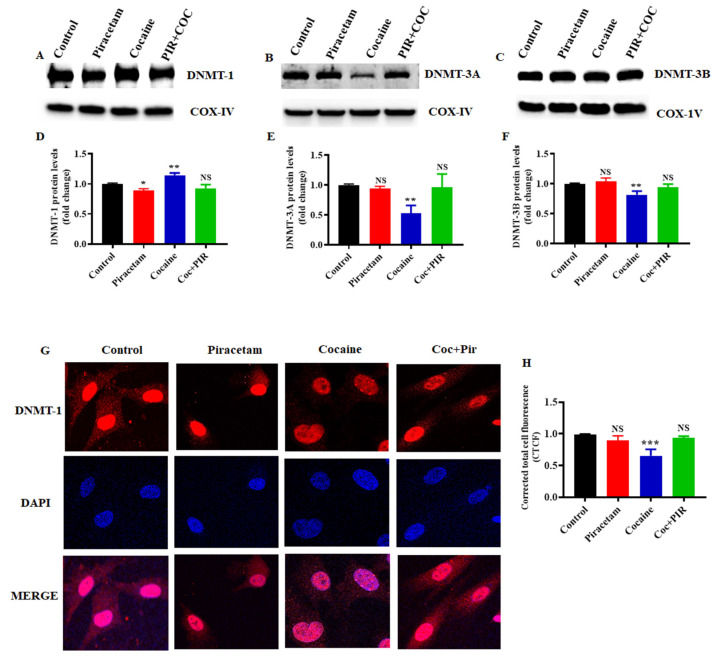
Piracetam reversed the impact of cocaine on DNMT protein expression in the mitochondrial fraction. Human primary astrocytes (5 × 10^6^ cells/mL) were exposed to cocaine (1 µM) and/or piracetam (10 µM) for 24 h. Mitochondrial fraction were resolved by SDS-PAGE and analyzed by western blots showing the expression of (**A**) DNMT-1, (**B**) DNMT-3A and (**C**) DNMT-3B. (**D**–**F**) Densitometric analysis of the level of each protein relative to the level of COX-IV as a loading control (fold change relative to the control). The data are expressed as the mean ± SD of three independent experiments. *** *p* < 0.001, ** *p* < 0.01, * *p* < 0.05, NS—nonsignificant. (**G**) DNMT-1 immunostaining (red) and cell nuclei, which were stained with DAPI (blue), were observed by confocal microscopy (magnification 100x, scale bar 100 μm). (**H**) Quantification of DNMT-1 fluorescence intensity (CTCF).

**Figure 4 brainsci-10-00611-f004:**
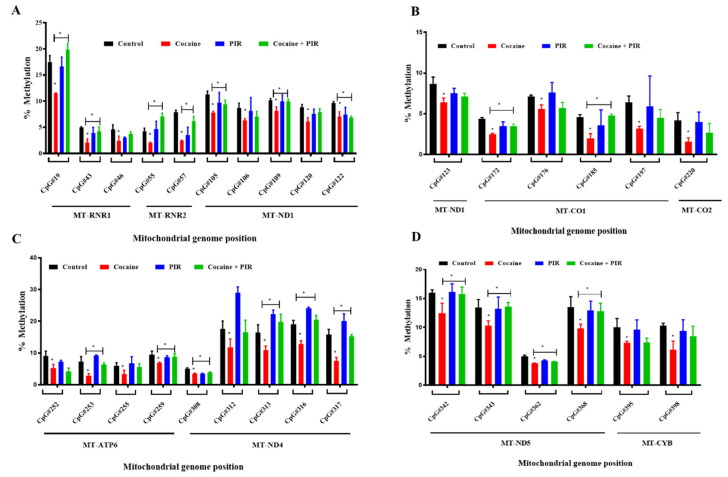
Analysis of mtDNA methylation by targeted next generation bisulfite sequencing (TNGBS). Human primary astrocytes were exposed to cocaine (1 µM) and/or piracetam (10 µM) for 24 h. The mtDNA methylation profiles were determined by TNGBS. Results represent the protective effect of piracetam on cocaine-induced hypomethylation in different mitochondrial CpG sites mt-RNR1, mt-RNR2, ND1, ND4, ND5, mt-CO1, mt-CO2, mt-ATP6 and mt-CYB (**A**–**D**). The results are expressed average of methylation percentage ± SD, * *p <* 0.05.

**Figure 5 brainsci-10-00611-f005:**
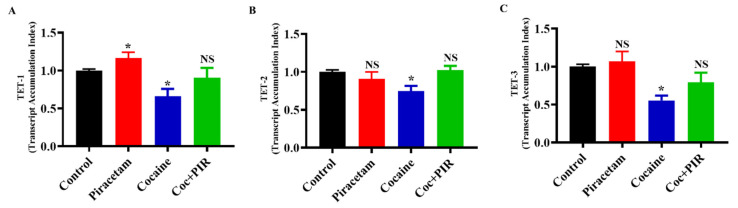
Piracetam reversed the impact of cocaine on TET gene expression. Human primary astrocytes (2 × 10^6^ cells/mL) were exposed to cocaine (1 µM) and/or piracetam (10 µM) for 24 h. The mRNA expression of (**A**) TET-1, (**B**) TET-2 and (**C**) TET-3 was determined by qRT-PCR analysis. The housekeeping gene β-actin was used as a loading control. The results are expressed as the mean ± SD of the TAI of three independent experiments. * *p <* 0.05, NS—nonsignificant.

**Figure 6 brainsci-10-00611-f006:**
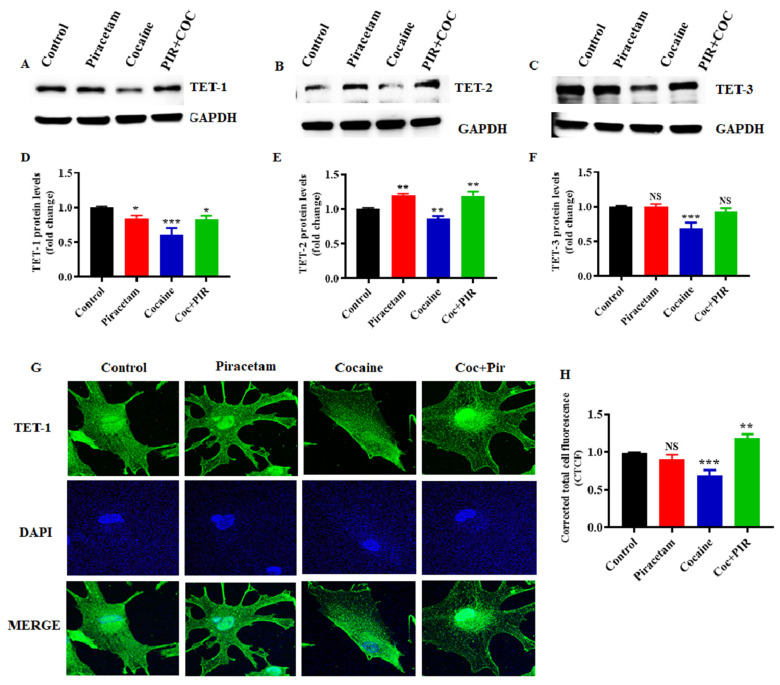
Piracetam reversed the impact of cocaine on TET protein expression. Human primary astrocytes (2.5 × 10^6^ cells/mL) were exposed to cocaine (1 µM) and/or piracetam (10 µM) for 24 h. Total cell lysates were resolved by SDS-PAGE and analyzed by western blots showing the expression of (**A**) TET-1, (**B**) TET-2 and (**C**) TET-3. (**D**–**F**) Densitometric analysis of the level of each protein relative to the level of GAPDH as a loading control (fold change relative to the control). The data are expressed as the mean ± SD of three independent experiments. *** *p <* 0.001, ** *p <* 0.01, * *p <* 0.05, NS—nonsignificant. (**G**) TET-1 (green) immunostaining and cell nuclei, which were stained with DAPI (blue), were observed by confocal microscopy (magnification 100× scale bar 100 μm). (**H**) Quantification of TET-1 fluorescence intensity (CTCF).

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
