# Peer review of "Neuroprotective Effect of Piracetam against Cocaine-Induced Neuro Epigenetic Modification of DNA Methylation in Astrocytes"

_brainsci, 2020, doi:10.3390/brainsci10090611_

Round 1

Reviewer 1 Report

This manuscript investigates the effects of Piracetam on astrocyte epigenetic modification following cocaine exposure in-vitro. While the exploration of astrocyte epigenetic modifications is timely I have reservations regarding the experimental design.

Line 80: Serum should not have been added to astrocyte media because it causes astrocyte reactivity. This is an important point to consider and implications should be addressed in the discussion.

Line 77-81: Cell density used and the day of culture on which the drugs where applied should be discussed in more detail.

Line 85-86: uM/ml? This unit measure makes no sense. Either it is uM or weight/ml not a combination of both otherwise you have uM/liter/ml. Please correct this throughout the manuscript.

Diluting drugs in water instead of media can distort the osmolarity of your cell culture. Further details need to be provided about the dilution of the drugs used and the volume added to the cell culture. If drugs were diluted in water and a significant volume of water was added to the cell culture then the implications on astrocyte cell culture health should be acknowledged and discussed.

Line 96: 'The percentage of methylated DNA was proportional to the measured intensity.' What does this mean? Please clarify in the text.

Figure 3G: Images are not clear. Please consider including higher magnification images to demonstrate what is described in lines 214 – 219.

Line 114: ‘astrocytes were grown…’ for how long? Please add additional experimental detail to the text.

Line 234 – 236: It is difficult to understand what is being conveyed in this sentence. I recommend rewording.

Figure 6H. Typo ‘Merage’ should say ‘Merge’

Line 313: This line uses the term ‘pretreatment’. This experimental paradigm was ‘co-treatment’, there was no pretreatment with Piracetam, since the two drugs were administered simultaneously. A different term should be used here.

Line 314: The phrase ‘restoring hypomethylation’ was used but based on the experimental paradigm ‘preventing hypomethylation’ would be more appropriate since drugs were administered simultaneously.

Line 342 and 356: In line with the previous comment ‘prevents’ not ‘reverses’ should be used in these sentences.

Somewhere in the text please comment on the translational relevance of the doses of Piracetam and cocaine that have been used in this study.

Author Response

Please find attached file_Reviewer Comments_Response_1

Reviewer 2 Report

In this manuscript, Sivalingam and Samikkannu show the effect of nootropic drug piracetam on DNA methylation levels after cocaine exposure. They used human primary cortical astrocytes to assed the effect of piracetam on cocaine-induced DNA methylation. Authors present interesting data that shows the involvement of mitochondrial DNMT’s in astrocytes but the rationale for analyzing mitochondrial DNMT’s is missing from the manuscript.

Line 102-104: Provide complete primer sequences used in qRT-PCR

Line 155-158: Which posthoc test was used to compare the groups? Were there any outliers? Which test was used to detect outliers? Did the authors remove these outliers before conducting ANOVA analysis?

Author should provide complete results of ANOVA in the format F(  ,    ) =          , p = .xxx

Line 189:  citation for this statement is missing “DNMT-1 activity is highly regulated…….”

Line 217-219: Authors should add the actual immunostaining quantification graph to figure 3 along with the images in support of these claims. How did the authors quantify immunostaining? Provide details in the methods section.

Line 227: Legend G mislabeled as H

Line 232-233: Provide details of the bisulfite sequencing method used for this analysis

Line 269-272: Similar to figure 3 results, authors should provide actual quantification of the immunostainings along with images in support of these claims. 

Figure 6: in panel-A, western blot image of Control group TET1 is impossible to quantify with such background, Author should replace this image. 

Figure 6: Immunostaining panel mislabeled as H instead of G.

Authors should provide full, unedited western blot images as supplementary materials.

Author Response

Please find attached file_Rev comments and Response_2

Round 2

Reviewer 1 Report

Reviewer comments and suggestions have been addressed.

Reviewer 2 Report

The authors have addressed all my suggestions. I found their responses satisfactory and the revised version has been much improved.